# NATURAL LANGUAGE INFERENCE OVER INTERACTION SPACE

**Yichen Gong**[†‡]**, Heng Luo**[‡]**, Jian Zhang**[‡]

[†] New York University, New York, USA
[‡] Horizon Robotics, Inc., Beijing, China
`yichen.gong@nyu.edu, {heng.luo, jian.zhang}@hobot.cc`

## ABSTRACT

Natural Language Inference (NLI) task requires an agent to determine the logical relationship between a natural language premise and a natural language hypothesis. We introduce Interactive Inference Network (IIN), a novel class of neural network architectures that is able to achieve high-level understanding of the sentence pair by hierarchically extracting semantic features from interaction space. We show that an interaction tensor (attention weight) contains semantic information to solve natural language inference, and a denser interaction tensor contains richer semantic information. One instance of such architecture, Densely Interactive Inference Network (DIIN), demonstrates the state-of-the-art performance on large scale NLI copora and large-scale NLI alike corpus. It's noteworthy that DIIN achieve a greater than 20% error reduction on the challenging Multi-Genre NLI (MultiNLI; Williams et al. 2017) dataset with respect to the strongest published system.

## 1 INTRODUCTION

Natural Language Inference (NLI also known as recognizing textual entiailment, or RTE) task requires one to determine whether the logical relationship between two sentences is among *entailment* (if the premise is true, then the hypothesis must be true), *contradiction* (if the premise is true, then the hypothesis must be false) and *neutral* (neither entailment nor contradiction). NLI is known as a fundamental and yet challenging task for natural language understanding(Williams et al., 2017), not only because it requires one to identify the language pattern, but also to understand certain common sense knowledge. In Table 1, three samples from MultiNLI corpus show solving the task requires one to handle the full complexity of lexical and compositional semantics. The previous work on NLI (or RTE) has extensively researched on conventional approaches(Fyodorov et al., 2000; Bos & Markert, 2005; MacCartney & Manning, 2009). Recent progress on NLI is enabled by the availability of 570k human annotated dataset(Bowman et al., 2015) and the advancement of representation learning technique.

Among the core representation learning techniques, attention mechanism is broadly applied in many NLU tasks since its introduction: machine translation(Bahdanau et al., 2014), abstractive summarization(Rush et al., 2015), Reading Comprehension(Hermann et al., 2015), dialog system(Mei et al., 2016), etc. As described by Vaswani et al. (2017), "*An attention function can be described as mapping a query and a set of key-value pairs to an output, where the query, keys, values, and output are all vectors. The output is computed as a weighted sum of the values, where the weight assigned to each value is computed by a compatibility function of the query with the corresponding key*". Attention mechanism is known for its alignment between representations, focusing one part of representation over another, and modeling the dependency regardless of sequence length. Observing attention's powerful capability, we hypothesize that the attention weight can assist with machine to understanding the text.

A regular attention weight, the core component of the attention mechanism, encodes the cross-sentence word relationship into a alignment matrix. However, a multi-head attention weightVaswani et al. (2017) can encode such interaction into multiple alignment matrices, which shows a more powerful alignment. In this work, we push the multi-head attention to a extreme by building a word-

| |
| --- |
| **Premise:** The FCC has created two tiers of small business for this service with the approval of the SBA. |
| **Hypothesis:** The SBA has given the go-ahead for the FCC to divide this service into two tiers of small business. |
| **Label:** entailment |
| **Premise:** He was crying like his mother had just walloped him. |
| **Hypothesis:** He was crying like his mother hit him with a spoon. |
| **Label:** Neutral |
| **Premise:** Later, Tom testified against John so as to avoid the electric chair. |
| **Hypothesis:** Tom refused to turn on his friend, even though he was slated to be executed. |
| **Label:** Contradiction |

Table 1: Samples from MultiNLI datasets.

by-word dimension-wise alignment tensor which we call interaction tensor. The interaction tensor encodes the high-order alignment relationship between sentences pair. Our experiments demonstrate that by capturing the rich semantic features in the interaction tensor, we are able to solve natural language inference task well, especially in cases with paraphrase, antonyms and overlapping words.

We dub the general framework as Interactive Inference Network(IIN). To the best of our knowledge, it is the first attempt to solve natural language inference task in the interaction space. We further explore one instance of Interactive Inference Network, Densely Interactive Inference Network (DIIN), which achieves new state-of-the-art performance on both SNLI and MultiNLI copora. To test the generality of the architecture, we interpret the paraphrase identification task as natural language inference task where matching as entailment, not-matching as neutral. We test the model on Quora Question Pair dataset, which contains over 400k real world question pair, and achieves new state-of-the-art performance.

We introduce the related work in Section 2, and discuss the general framework of IIN along with a specific instance that enjoys state-of-the-art performance on multiple datasets in Section 3. We describe experiments and analysis in Section 4. Finally, we conclude and discuss future work in Section 5.

## 2 RELATED WORK

The early exploration on NLI mainly rely on conventional methods and small scale datasets(Marelli et al., 2014). The availability of SNLI dataset with 570k human annotated sentence pairs has enabled a good deal of progress on natural language understanding. The essential representation learning techniques for NLU such as attention(Wang & Jiang, 2015), memory(Munkhdalai & Yu, 2016) and the use of parse structure(Bowman et al., 2016; Mou et al., 2015) are studied on the SNLI which serves as an important benchmark for sentence understanding. The models trained on NLI task can be divided into two categories: (i) sentence encoding-based model which aims to find vector representation for each sentence and classifies the relation by using the concatenation of two vector representation along with their absolute element-wise difference and element-wise product(Bowman et al., 2016; Vendrov et al., 2015; Mou et al., 2015; Liu et al., 2016; Munkhdalai & Yu, 2016). (ii) Joint feature models which use the cross sentence feature or attention from one sentence to another(Rocktäschel et al., 2015; Wang & Jiang, 2015; Cheng et al., 2016; Parikh et al., 2016; Wang et al., 2017; Yu & Munkhdalai, 2017; Sha et al., 2016).

After neural attention mechanism is successfully applied on the machine translation task, such technique has became widely used in both natural language process and computer vision domains. Many variants of attention technique such as hard-attention(Xu et al., 2015), self-attention(Parikh et al., 2016), multi-hop attention(Gong & Bowman, 2017), bidirectional attention(Seo et al., 2016) and multi-head attention(Vaswani et al., 2017) are also introduced to tackle more complicated tasks. Before this work, neural attention mechanism is mainly used to make alignment, focusing on specific part of the representation. In this work, we want to show that attention weight contains rich semantic information required for understanding the logical relationship between sentence pair.

Though RNN or LSTM are very good for variable length sequence modeling, using Convolutional neural network in NLU tasks is very desirable because of its parallelism in computation. Convolutional structure has been successfully applied in various domain such as machine translation(Gehring et al., 2017), sentence classification(Kim, 2014), text matching(Hu et al., 2014) and sentiment analysis(Kalchbrenner et al., 2014), etc. The convolution structure is also applied on different level of granularity such as byte(Zhang & LeCun, 2017), character(Zhang et al., 2015), word(Gehring et al., 2017) and sentences(Mou et al., 2015) levels.

## 3 MODEL

### 3.1 INTERACTIVE INFERENCE NETWORK

The Interactive Inference Network (IIN) is a hierarchical multi-stage process and consists of five components. Each of the components is compatible with different type of implementations. Potentially all exiting approaches in machine learning, such as decision tree, support vector machine and neural network approach, can be transfer to replace certain component in this architecture. We focus on neural network approaches below. Figure 1 provides a visual illustration of Interactive Inference Network.

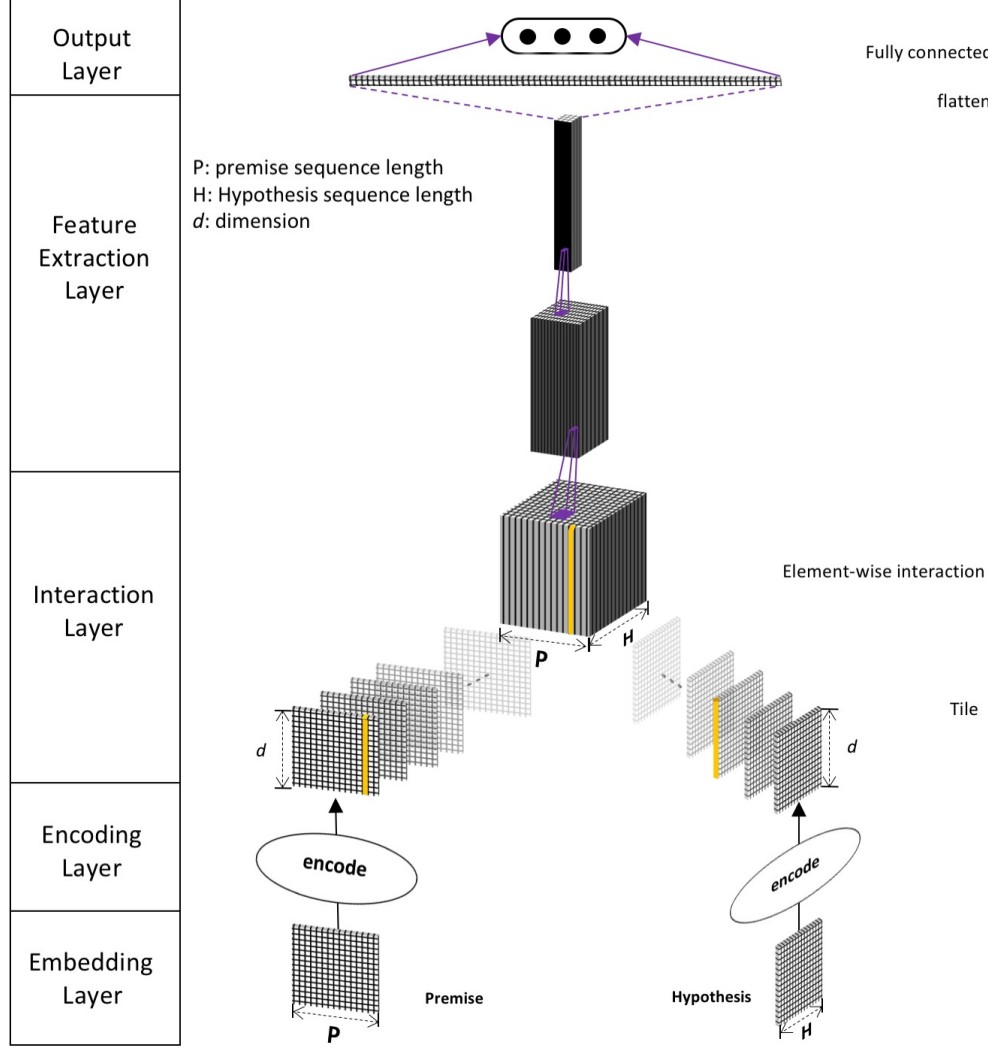

Figure 1: A visual illustration of Interactive Inference Network (IIN).

1. **Embedding Layer** converts each word or phrase to a vector representation and construct the representation matrix for sentences. In embedding layer, a model can map tokens to vectors with the pre-trained word representation such as GloVe(Pennington et al., 2014), word2Vec(Mikolov et al., 2013) and fasttext(Joulin et al., 2016). It can also utilize the pre-processing tool, e.g. named entity recognizer, part-of-speech recognizer, lexical parser and coreference identifier etc., to incorporate more lexical and syntactical information into the feature vector.

2. **Encoding Layer** encodes the representations by incorporating the context information or enriching the representation with desirable features for future use. For instance, a model can adopt bidirectional recurrent neural network to model the temporal interaction on both direction, recursive neural network(Socher et al., 2011) (also known as TreeRNN) to model the compositionality and the recursive structure of language, or self-attention to model the long-term dependency on sentence. Different components of encoder can be combined to obtain a better sentence matrix representation.

3. **Interaction Layer** creates an word-by-word interaction tensor by both premise and hypothesis representation matrix. The interaction can be modeled in different ways. A common approach is to compute the cosine similarity or dot product between each pair of feature vector. On the other hand, a high-order interaction tensor can be constructed with the outer product between two matrix representations.

4. **Feature Extraction Layer** adopts feature extractor to extract the semantic feature from interaction tensor. The convolutional feature extractors, such as AlexNet(Krizhevsky et al., 2012), VGG(Simonyan & Zisserman, 2014), Inception(Szegedy et al., 2014), ResNet(He et al., 2016) and DenseNet(Huang et al., 2016), proven work well on image recognition are completely compatible under such architecture. Unlike the work (Kim, 2014; Zhang et al., 2015) who employs 1-D sliding window, our CNN architecture allows 2-D kernel to extract semantic interaction feature from the word-by-word interaction between n-gram pair. Sequential or tree-like feature extractors are also applicable in the feature extraction layer.

5. **Output Layer** decodes the acquired features to give prediction. Under the setting of NLI, the output layer predicts the confidence on each class.

## 3.2 DENSELY INTERACTIVE INFERENCE NETWORK

Here we introduce Densely Interactive Inference Network (DIIN)[1], which is a relatively simple instantiation of IIN but produces state-of-the-art performance on multiple datasets.

**Embedding Layer:** For DIIN, we use the concatenation of word embedding, character feature and syntactical features. The word embedding is obtained by mapping token to high dimensional vector space by pre-trained word vector (840B GloVe). The word embedding is updated during training. As in (Kim et al., 2016; Lee et al., 2016), we filter character embedding with 1D convolution kernel. The character convolutional feature maps are then max pooled over time dimension for each token to obtain a vector. The character features supplies extra information for some out-of-vocabulary (OOV) words. Syntactical features include one-hot part-of-speech (POS) tagging feature and binary exact match (EM) feature. The EM value is activated if there are tokens with same stem or lemma in the other sentence as the corresponding token. The EM feature is simple while found useful as in reading comprehension task (Chen et al., 2017a). In analysis section, we study how EM feature helps text understanding. Now we have premise representation $P \in \mathbb{R}^{p \times d}$ and hypothesis representation $H \in \mathbb{R}^{h \times d}$, where $p$ refers to the sequence length of premise, $h$ refers to the sequence length of hypothesis and $d$ means the dimension of both representation. The 1-D convolutional neural network and character features weights share the same set of parameters between premise and hypothesis.

**Encoding Layer:** In the encoding layer, the premise representation $P$ and the hypothesis representation $H$ are passed through a two-layer highway network, thus having $\hat{P} \in \mathbb{R}^{p \times d}$ and

---

[1]The code is open sourced at https://github.com/YichenGong/Densely-Interactive-Inference-Network

$\hat{\boldsymbol{H}} \in \mathbb{R}^{h \times d}$ for new premise representation and new hypothesis representation. These new representation are then passed to self-attention layer to take into account the word order and context information. Take premise as example, we model self-attention by

$$\boldsymbol{A}_{ij} = \alpha(\hat{\boldsymbol{P}}_i), \hat{\boldsymbol{P}}_j \in \mathbb{R} \tag{1}$$

$$\bar{\boldsymbol{P}}_i = \sum_{j=1}^{p} \frac{\exp(\boldsymbol{A}_{ij})}{\sum_{k=1}^{p} \exp(\boldsymbol{A}_{kj})} \hat{\boldsymbol{P}}_j, \tag{2}$$
$$\forall i, j \in [1, ..., p]$$

where $\bar{\boldsymbol{P}}_i$ is a weighted summation of $\hat{\boldsymbol{P}}$. We choose $\alpha(\boldsymbol{a}, \boldsymbol{b}) = \boldsymbol{w}_a^\top [\boldsymbol{a}; \boldsymbol{b}; \boldsymbol{a} \circ \boldsymbol{b}]$, where $\boldsymbol{w}_a \in \mathbb{R}^{3d}$ is a trainable weight, $\circ$ is element-wise multiplication, [;] is vector concatenation across row, and the implicit multiplication is matrix multiplication. Then both $\hat{\boldsymbol{P}}$ and $\bar{\boldsymbol{P}}$ are fed into a semantic composite fuse gate (fuse gate in short), which acts as a skip connection. The fuse gate is implemented as

$$\boldsymbol{z}_i = \tanh(\boldsymbol{W}^{1\top}[\hat{\boldsymbol{P}}_i; \bar{\boldsymbol{P}}_i] + \boldsymbol{b}^1) \tag{3}$$
$$\boldsymbol{r}_i = \sigma(\boldsymbol{W}^{2\top}[\hat{\boldsymbol{P}}_i; \bar{\boldsymbol{P}}_i] + \boldsymbol{b}^2) \tag{4}$$
$$\boldsymbol{f}_i = \sigma(\boldsymbol{W}^{3\top}[\hat{\boldsymbol{P}}_i; \bar{\boldsymbol{P}}_i] + \boldsymbol{b}^3) \tag{5}$$
$$\tilde{\boldsymbol{P}}_i = \boldsymbol{r}_i \circ \hat{\boldsymbol{P}}_i + \boldsymbol{f}_i \circ \boldsymbol{z}_i \tag{6}$$

where $\boldsymbol{W}^1, \boldsymbol{W}^2, \boldsymbol{W}^3 \in \mathbb{R}^{2d \times d}$ and $\boldsymbol{b}^1\ \boldsymbol{b}^2, \boldsymbol{b}^3 \in \mathbb{R}^d$ are trainable weights, $\sigma$ is sigmoid nonlinear operation.

We do the same operation on hypothesis representation, thus having $\tilde{\boldsymbol{H}}$. The weights of intra-attention and fuse gate for premise and hypothesis are not shared, but the difference between the weights of are penalized. The penalization aims to ensure the parallel structure learns the similar functionality but is aware of the subtle semantic difference between premise and hypothesis.

**Interaction Layer:** The interaction layer models the interaction between premise encoded representation $\boldsymbol{P}^{enc}$ and hypothesis encoded representation $\boldsymbol{H}^{enc}$ as follows:

$$\boldsymbol{I}_{ij} = \beta(\tilde{\boldsymbol{P}}_i, \tilde{\boldsymbol{H}}_j) \in \mathbb{R}^d, \forall i \in [1, ..., p], \forall j \in [1, ..., h] \tag{7}$$

where $\tilde{\boldsymbol{P}}_i$ is the $i$-th row vector of $\tilde{\boldsymbol{P}}$, and $\tilde{\boldsymbol{H}}_j$ is the $j$-th row vector of $\tilde{\boldsymbol{H}}$. Though there are many implementations of interaction, we find $\beta(a, b) = a \circ b$ very useful.

**Feature Extraction Layer:** We adopt DenseNet(Huang et al., 2016) as convolutional feature extractor in DIIN. Though our experiments show ResNet(He et al., 2016) works well in the architecture, we choose DenseNet because it is effective in saving parameters. One interesting observation with ResNet is that if we remove the skip connection in residual structure, the model does not converge at all. We found batch normalization delays convergence without contributing to accuracy, therefore we does not use it in our case. A ReLU activation function is applied after all convolution unless otherwise noted. Once we have the interaction tensor $\boldsymbol{I}$, we use a convolution with $1 \times 1$ kernel to scale down the tensor in a ratio, $\eta$, without following ReLU. If the input channel is $k$ then the output channel is $floor(k \times \eta)$. Then the generated feature map is feed into three sets of Dense block(Huang et al., 2016) and transition block pair. The DenseNet block contains $n$ layers of $3 \times 3$ convolution layer with growth rate of $g$. The transition layer has a convolution layer with $1 \times 1$ kernel for scaling down purpose, followed by a max pooling layer with stride 2. The transition scale down ratio in transition layer is $\theta$.

**Output Layer:** DIIN uses a linear layer to classify final flattened feature representation to three classes.

## 4 EXPERIMENTS

In this section, we present the evaluation of our model. We first perform quantitative evaluation, comparing our model with other competitive models. We then conduct some qualitative analyses to understand how DIIN achieve the high level understanding through interaction.

### 4.1 DATA

Here we introduce three datasets we evaluate our model on. The evaluation metric for all dataset is accuracy.

**SNLI**  Stanford Natural Language Inference (SNLI; Bowman et al. 2015) has 570k human anno­tated sentence pairs. The premise data is draw from the captions of the Flickr30k corpus, and the hypothesis data is manually composed. The labels provided in are "entailment", "neutral', "contra­diction" and "-". "-" shows that annotators cannot reach consensus with each other, thus removed during training and testing as in other works. We use the same data split as in Bowman et al. (2015).

**MultiNLI**  Multi-Genre NLI Corpus (MultiNLI; Williams et al. 2017) has 433k sentence pairs, whose collection process and task detail are modeled closely to SNLI. The premise data is col­lected from maximally broad range of genre of American English such as written non-fiction genres (SLATE, OUP, GOVERNMENT, VERBATIM, TRAVEL), spoken genres (TELEPHONE, FACE-TO-FACE), less formal written genres (FICTION, LETTERS) and a specialized one for 9/11. Half of these selected genres appear in training set while the rest are not, creating in-domain (matched) and cross-domain (mismatched) development/test sets. We use the same data split as provided by Williams et al. (2017). Since test set labels are not provided, the test performance is obtained through submission on Kaggle.com[2]. Each team is limited to two submissions per day.

**Quora question pair**  Quora question pair dataset contains over 400k real world question pair selected from Quora.com. A binary annotation which stands for *match (duplicate)* or *not match (not duplicate)* is provided for each question pair. In our case, *duplicate* question pair can be interpreted as *entailment* relation and *not duplicate* as *neutral*. We use the same split ratio as mentioned in (Wang et al., 2017).

### 4.2 EXPERIMENTS SETTING

We implement our algorithm with Tensorflow(Abadi et al., 2016) framework. An Adadelta opti­mizer(Zeiler, 2012) with $\rho$ as 0.95 and $\epsilon$ as $1e-8$ is used to optimize all the trainable weights. The initial learning rate is set to 0.5 and batch size to 70. When the model does not improve best in-domain performance for 30,000 steps, an SGD optimizer with learning rate of $3e-4$ is used to help model to find a better local optimum. Dropout layers are applied before all linear layers and after word-embedding layer. We use an exponential decayed keep rate during training, where the initial keep rate is 1.0 and the decay rate is 0.977 for every 10,000 step. We initialize our word embeddings with pre-trained 300D GloVe 840B vectors Pennington et al. (2014) while the out-of-vocabulary word are randomly initialized with uniform distribution. The character embeddings are randomly initialized with 100D. We crop or pad each token to have 16 characters. The 1D convolution ker­nel size for character embedding is 5. All weights are constraint by L2 regularization, and the L2 regularization at step $t$ is calculated as follows:

$$L2Ratio_t = \sigma(\frac{(t - L2FullStep/2) \times 8}{L2FullStep/2}) \times L2FullRatio \tag{8}$$

where $L2FullRatio$ determines the maximum L2 regularization ratio, and $L2FullStep$ determines at which step the maximum L2 regularization ratio would be applied on the L2 regularization. We choose $L2FullRatio$ as $0.9e-5$ and $L2FullStep$ as 100,000. The ratio of L2 penalty between the

---

[2]In-domain (matched) leaderboard: `https://inclass.kaggle.com/c/multinli-matched-open-evaluation/leaderboard`; cross-domain(mismatched) leader­board: `https://inclass.kaggle.com/c/multinli-mismatched-open-evaluation/leaderboard`

| Model | Test Accuracy | |
| --- | --- | --- |
| | **Matched** | **Mismatched** |
| 1. BiLSTM(Williams et al., 2017) | 67.0 | 67.6 |
| 2. InnerAtt(Balazs et al., 2017) | 72.1 | 72.1 |
| 3. ESIM(Williams et al., 2017) | 72.3 | 72.1 |
| 4. Gated-Att BiLSTM(Chen et al., 2017b) | 73.2 | 73.6 |
| 5. Shorcut-Stacked encoder(Nie & Bansal, 2017) | 74.6 | 73.6 |
| 6. DIIN | **78.8** | **77.8** |
| 7. InnerAtt(ensemble) | 72.2 | 72.8 |
| 8. Gated-Att BiLSTM (ensemble) | 74.9 | 74.9 |
| 9. DIIN (ensemble) | **80.0** | **78.7** |

Table 2: MultiNLI result.

difference of two encoder weights is set to $1e-3$. For a dense block in feature extraction layer, the number of layer $n$ is set to $8$ and growth rate $g$ is set to $20$. The first scale down ratio $\eta$ in feature extraction layer is set to $0.3$ and transitional scale down ratio $\theta$ is set to $0.5$. The sequence length is set as a hard cutoff on all experiments: 48 for MultiNLI, 32 for SNLI and 24 for Quora Question Pair Dataset. During the experiments on MultiNLI, we use 15% of data from SNLI as in Williams et al. (2017). We select the parameter by the best run of development accuracy. Our ensembling approach considers the majority vote of the predictions given by multiple runs of the same model under different random parameter initialization.

### 4.3 EXPERIMENT ON MULTINLI

We compare our result with all other published systems in Table 2. Besides ESIM, the state-of-the-art model on SNLI, all other models appear at RepEval 2017 workshop. RepEval 2017 workshop requires all submitted model to be sentence encoding-based model therefore alignment between sentences and memory module are not eligible for competition. All models except ours share one common feature that they use LSTM as a essential building block as encoder. Our approach, without using any recurrent structure, achieves the new state-of-the-art performance of 80.0%, exceeding current state-of-the-art performance by more than 5%. Unlike the observation from Nangia et al. (2017), we find the out-of-domain test performance is consistently lower than in-domain test performance. Selecting parameters from the best in-domain development accuracy partially contributes to this result.

### 4.4 EXPERIMENT ON SNLI

In Table 3, we compare our model to other model performance on SNLI. Experiments (2-7) are sentence encoding based model. Bowman et al. (2016) provides a BiLSTM baseline. Vendrov et al. (2015) adopts two layer GRU encoder with pre-trained "skip-thoughts" vectors. To capture sentence-level semantics, Mou et al. (2015) use tree-based CNN and Bowman et al. (2016) propose a stack-augmented parser-interpreter neural network (SPINN) which incorporates parsing information in a sequential manner. Liu et al. (2016) uses intra-attention on top of BiLSTM to generate sentence representation, and Munkhdalai & Yu (2016) proposes an memory augmented neural network to encode the sentence. The next group of model, experiments (8-18), uses cross sentence feature. Rocktäschel et al. (2015) aligns each sentence word-by-word with attention on top of LSTMs. Wang & Jiang (2015) enforces cross sentence attention word-by-word matching with the proprosed mLSTM model. Cheng et al. (2016) proposes long short-term memory-network(LSTMN) with deep attention fusion that links the current word to previous word stored in memory. Parikh et al. (2016) decomposes the task into sub-problems and conquer them respectively. Yu & Munkhdalai (2017) proposes neural tree indexer, a *full n-ary tree* whose subtrees can be overlapped. Re-read LSTM proposed by Sha et al. (2016) considers the attention vector of one sentence as the inner-state of LSTM for another sentence. Chen et al. (2016) propose a sequential model that infers locally, and a ensemble with tree-like inference module that further improves performance. We show our model, DIIN, achieves state-of-the-art performance on the competitive leaderboard.

| Model | Test Accuracy SNLI |
|---|---|
| 1. Handcrafted features(Bowman et al., 2015) | 78.2 |
| 2. LSTM encoder(Bowman et al., 2016) | 80.6 |
| 3. pretrained GRU encoders(Vendrov et al., 2015) | 81.4 |
| 4. tree-based CNN encoders(Mou et al., 2015) | 82.1 |
| 5. SPINN-PI encoders(Bowman et al., 2016) | 83.2 |
| 6. BiLSTM intra-attention encoders(Liu et al., 2016) | 84.2 |
| 7. NSE encoders(Munkhdalai & Yu, 2016) | 84.6 |
| 8. LSTM with attention(Rocktäschel et al., 2015) | 83.5 |
| 9. mLSTM(Wang & Jiang, 2015) | 86.1 |
| 10. LSTMN with deep attention fusion(Cheng et al., 2016) | 86.3 |
| 11. decomposable attention model(Parikh et al., 2016) | 86.3 |
| 12. Intra-sentence attention + (11)(Parikh et al., 2016) | 86.8 |
| 13. BiMPM(Wang et al., 2017) | 86.9 |
| 14. NTI-SLSTM-LSTM(Yu & Munkhdalai, 2017) | 87.3 |
| 15. re-read LSTM(Sha et al., 2016) | 87.5 |
| 16. ESIM(Chen et al., 2016) | 88.0 |
| 17. ESIM ensemble with syntactic tree-LSTM(Chen et al., 2016) | 88.6 |
| 18. BiMPM (ensemble)(Wang et al., 2017) | 88.8 |
| 19. DIIN | **88.0** |
| 20. DIIN (ensemble) | **88.9** |

Table 3: SNLI result.

| Model | Accuracy | |
|---|---|---|
| | Dev Acc | Test Acc |
| 1. Siamese-CNN | - | 79.60 |
| 2. Multi-Perspective CNN | - | 81.38 |
| 3. Siamese-LSTM | - | 82.58 |
| 4. Multi-Perspective-LSTM | - | 83.21 |
| 5. L.D.C | - | 85.55 |
| 6. BiMPM(Wang et al., 2017) | 88.69 | 88.17 |
| 7. pt-$\text{DECATT}_{word}$(Tomar et al., 2017) | 88.44 | 87.54 |
| 8. pt-$\text{DECATT}_{char}$(Tomar et al., 2017) | 88.89 | 88.40 |
| 9. DIIN | 89.44 | 89.06 |
| 10. DIIN (ensemble) | **90.48** | **89.84** |

Table 4: Quora question dataset result. First six rows are copied from Wang et al. (2017) and next two rows from (Tomar et al., 2017).

## 4.5 EXPERIMENT ON QUORA QUESTION PAIR DATASET

In this subsection, we evaluate the effectiveness of our model for paraphrase identification as natural language inference task. Other than our baselines, we compare with Wang et al. (2017) and Tomar et al. (2017). BIMPM models different perspective of matching between sentence pair on both direction, then aggregates matching vector with LSTM. $\text{DECATT}_{word}$ and $\text{DECATT}_{char}$ uses automatically collected in-domain paraphrase data to noisy pretrain $n$-gram word embedding and $n$-gram subword embedding correspondingly on decomposable attention model proposed by (Parikh et al., 2016). In Table 4, our experiment shows DIIN has better performance than all other models and an ensemble score is higher than the former best result for more than 1 percent.

## 4.6 ANALYSIS

**Ablation Study** We conduct a ablation study on our base model to examine the effectiveness of each component. We study our model on MultiNLI dataset and we use Matched validation score as the standard for model selection. The result is shown in Table 5. We studies how EM feature

| Ablation Experiments | Dev Accuracy | |
|---|---|---|
| | Matched | Mismatched |
| 1. DIIN | 79.2 | 79.1 |
| 2. DIIN - EM feature | 78.2 | 78.0 |
| 3. DIIN - conv structure | 73.2 | 73.6 |
| 4. DIIN - encoding layer | 73.5 | 73.2 |
| 5. DIIN - self-att and fuse gate | 77.7 | 77.3 |
| 6. DIIN - fuse gate | 73.5 | 73.8 |
| 7. DIIN - fuse gate + addition as skip connection | 77.3 | 76.3 |
| 8. DIIN - dense interaction tensor + similarity matrix | 75.2 | 75.5 |
| 9. DIIN - tied encoding layer parameter | 78.5 | 78.3 |

Table 5: Ablation study result.

contributes to the system. After removing the exact match binary feature, we find the performance degrade to 78.2 on matched score on development set and 78.0 on mismatched score. As observed in reading comprehension task Chen et al. (2017a), the simple exact match feature does help the model to better understand the sentences. In the experiment 3, we remove the convolutional feature extractor and then model is structured as a sentence-encoding based model. The sentence representation matrix is max-pooled over time to obtain a feature vector. Once we have the feature vector $p$ for premise and $h$ for hypothesis, we use $[p; h; |p - h|; p \circ h]$ as final feature vector to classify the relationship. We obtain 73.2 for matched score and 73.6 on mismatched data. The result is competitive among other sentence-encoding based model. We further study how encoding layer contribute in enriching the feature space in interaction tensor. If we remove encoding layer completely, then we'll obtain a 73.5 for matched score and 73.2 for mismatched score. The result demonstrate the feature extraction layer have powerful capability to capture the semantic feature. In experiment 5, we remove both self-attention and fuse gate, thus retaining only highway network. The result improves to 77.7 and 77.3 respectively on matched and mismatched development set. However, in experiment 6, when we only remove fuse gate, to our surprise, the performance degrade to 73.5 for matched score and 73.8 for mismatched. On the other hand, if we use the addition of the representation after highway network and the representation after self-attention as skip connection as in experiment 7, the performance increase to 77.3 and 76.3. The comparison indicates self-attention layer makes the training harder to converge while a skip connection could ease the gradient flow for both highway layer and self-attention layer. By comparing the base model and the model the in experiment 6, we show that the fuse gate not only well serves as a skip connection, but also makes good decision upon which information the fuse for both representation. To show that dense interaction tensor contains more semantic information, we replace the dense interaction tensor with dot product similarity matrix between the encoded representation of premise and hypothesis. The result shows that the dot product similarity matrix has an inferior capacity of semantic information. Another dimensionality study is provided in supplementary material. In experiment 9, we share the encoding layer weight, and the result decrease from the baseline. The result shows that the two set of encoding weights learn the subtle difference between premise and hypothesis.

**Error analysis** To analyze the model prediction, we use annotated subset of development set provided by Williams et al. (2017) that consists of 1,000 examples each tagged with zero or more following tags:

- CONDITIONAL: whether the sentence contains a conditional.
- WORD OVERLAP: whether both sentences share more than 70% of their tokens.
- NEGATION: whether a negation shows up in either sentence.
- ANTO: whether two sentences contain antonym pair.
- LONG SENTENCE: whether premise or hypothesis is longer than 30 or 16 tokens respectively.
- TENSE DIFFERENCE: whether any verb in two sentences uses different tense.
- ACTIVE/PASSIVE: whether there is an active-to-passive (or vice versa) transformation from the premise to the hypothesis.

| | Annotation Tag | Label Frequency | BiLSTM | BaLazs | Chen | DIIN |
|---|---|---|---|---|---|---|
| Matched | CONDITIONAL | 5% | 100% | 100% | 100% | 57% |
| | WORD OVERLAP | 6% | 50% | 63% | 63% | **79**% |
| | NEGATION | 26% | 71% | 75% | 75% | 78% |
| | ANTO | 3% | 67% | 50% | 50% | **82**% |
| | LONG SENTENCE | 20% | 50% | 75% | 67% | **81**% |
| | TENSE DIFFERENCE | 10% | 64% | 68% | 86% | 84% |
| | ACTIVE/PASSIVE | 3% | 75% | 75% | 88% | 93% |
| | PARAPHRASE | 5% | 78% | 83% | 78% | **88**% |
| | QUANTITY/TIME REASONING | 3% | 50% | 50% | 33% | 53% |
| | COREF | 6% | 83% | 83% | 83% | 77% |
| | QUANTIFIER | 25% | 64% | 59% | 74% | 74% |
| | MODAL | 29% | 66% | 65% | 75% | 84% |
| | BELIEF | 13% | 74% | 71% | 73% | **77**% |
| Mismatched | CONDITIONAL | 5% | 100% | 80% | 100% | 69% |
| | WORD OVERLAP | 7% | 58% | 62% | 76% | **92**% |
| | NEGATION | 21% | 69% | 73% | 72% | 77% |
| | ANTO | 4% | 58% | 58% | 58% | **80**% |
| | LONG SENTENCE | 20% | 55% | 67% | 67% | **73**% |
| | TENSE DIFFERENCE | 4% | 71% | 71% | 89% | 78% |
| | ACTIVE/PASSIVE | 2% | 82% | 82% | 91% | 70% |
| | PARAPHRASE | 7% | 81% | 89% | 89% | **100**% |
| | QUANTITY/TIME REASONING | 8% | 46% | 54% | 46% | 69% |
| | COREF | 6% | 80% | 70% | 80% | 79% |
| | QUANTIFIER | 28% | 70% | 68% | 77% | 78% |
| | MODAL | 25% | 67% | 67% | 76% | 75% |
| | BELIEF | 12% | 73% | 71% | 74% | **81**% |

Table 6: MultiNLI result.

- PARAPHRASE: whether the two sentences are close paraphrases
- QUANTITY/TIME REASONING: whether understanding the pair requires quantity or time reasoning.
- COREF: Whether the hypothesis contains a pronoun or referring expression that needs to be resolved using the premise.
- QUANTIFIER: Whether either sentence contains one of the following quantifier: *much, enough, more, most, less, least, no, none, some, any, many, few, several, almost, nearly.*
- MODAL: Whether one of the following modal verbs appears in either sentence: *can, could, may, might, must, will, would, should.*
- BELIEF: Whether one of the following belief verbs appear in either sentence: *know, believe, understand, doubt, think, suppose, recognize, forget, remember, imagine, mean, agree, disagree, deny, promise.*

For more detailed descriptions, please resort to Williams et al. (2017). The result is shown in Table 6. We find DIIN is consistently better on sentence pair with WORD OVERLAP, ANTO, LONG SENTENCE, PARAPHRASE and BELIEF tags by a large margin. During investigation, we hypothesize exact match feature helps the model to better understand paraphrase, therefore we study the result from second ablation ablation study where exact match feature is not used. Surprisingly, the model without exact model feature does not work worse on PARAPHRASE, instead, the accuracy on ANTO drops about 10%. DIIN is also work well on LONG SENTENCE, partially because the receptive field is large enough to cover all tokens.

**Visualization** We also visualize the hidden representation from interaction tensor $I$ and the feature map from first dense block in Figure 2. We pick a sentence pair whose premise is "*South Carolina has no referendum right, so the Supreme Court canceled the vote and upheld the ban.*" and hypothesis is "*South Carolina has a referendum right, so the Supreme Court was powerless over the state.*". The upper row of figures are sampled from hidden representation of interaction tensor $I$. We observe the values of neurons are highly correlated row-wise and column-wise in the interaction tensor $I$ and different channel of hidden representation shows different aspect of interaction. Though in certain channel same words, "*referendum*", or phrases, "*supreme court*", cause activation, different word or phrase pair, such as "*ban*" and "*powerless over*", also cause activation in other activation. It shows the model's strong capacity of understanding text in different perspective. The lower row of Figure 2 shows the feature map from first dense block. After being convolved from the interaction tensor

and previous feature map, new feature maps shows activation in different position, demonstrating different semantic features are found. The first figure in the lower row has similar pattern as normal attention weight whereas others has no obvious pattern. Different channels of feature maps indicate different kinds of semantic feature.

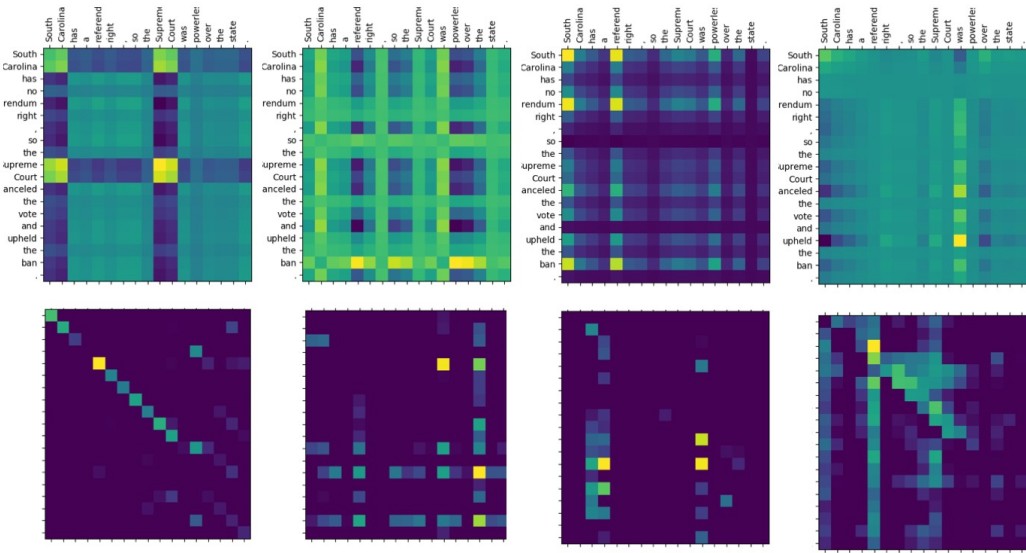

Figure 2: A visualization of hidden representation. The premise is "*South Carolina has no referendum right, so the Supreme Court canceled the vote and upheld the ban.*" and the hypothesis is "*South Carolina has a referendum right, so the Supreme Court was powerless over the state.*". The upper row are sampled from interaction tensor $I$ and the lower row are sample from the feature map of first dense block. We use viridis colormap, where yellow represents activation and purple shows the neuron is not active.

## 5    CONCLUSION AND FUTURE WORK

We show the interaction tensor (or attention weight) contains semantic information to understand the natural language. We introduce Interactive Inference Network, a novel class of architecture that allows the model to solve NLI or NLI alike tasks via extracting semantic feature from interaction tensor end-to-end. One instance of such architecture, Densely Interactive Inference Network (DIIN), achieves state-of-the-art performance on multiple datasets. By ablating each component in DIIN and changing the dimensionality, we show the effectiveness of each component in DIIN.

Though we have the initial exploration of natural language inference in interaction space, the full potential is not yet clear. We will keep exploring the potential of interaction space. Incorporating common-sense knowledge from external resources such as knowledge base to leverage the capacity of the mode is another research goal of ours.

ACKNOWLEDGMENTS

We thank Yuchen Lu, Chang Huang and Kai Yu for their sincere and insightful advice.

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

## A  SUPPLEMENTARY MATERIAL

**Dimensionality and Parameter number study**    To study the influence of the model dimension $d$ which is also the channel number of interaction tensor, we design experiments to find out whether dimension has influence on performance. We also present the parameter count of these models. The dimensionality is 448 where 300 comes from word embedding, 100 comes from char feature, 47 comes from Part of speech tagging and 1 comes from the binary exact match feature. Since Highway network sets the output dimensionality default as that in input, we design a variant to highway network so that different output size could be obtained. The variant of highway layer is designed as follows:

$$\boldsymbol{t}_i = \tanh(\boldsymbol{W}_t^\top \boldsymbol{x}_i + \boldsymbol{b}_t) \tag{9}$$

$$\boldsymbol{g}_i = \sigma(\boldsymbol{W}_g^\top \boldsymbol{x}_i + \boldsymbol{b}_g) \tag{10}$$

$$\boldsymbol{x}_i' = \begin{cases} \boldsymbol{x}_i & d_{in} = d_{out} \\ \boldsymbol{W}_x^\top \boldsymbol{x}_i + \boldsymbol{b}_x & d_{in} \neq d_{out} \end{cases} \tag{11}$$

$$\boldsymbol{o}_i = \boldsymbol{g}_i \circ \boldsymbol{t}_i + (1 - \boldsymbol{g}_i) \circ \boldsymbol{x}_i' \tag{12}$$

where $\boldsymbol{x}_i$ is the $i$-th vector of input matrix $\boldsymbol{x}$, $\boldsymbol{o}_i$ is the $i$-th vector of output matrix $\boldsymbol{o}$, $\boldsymbol{W}_t^\top$, $\boldsymbol{W}_g^\top$, $\boldsymbol{W}_x^\top \in \mathbb{R}^{d_{in} \times d_{out}}$ and $\boldsymbol{b}_t, \boldsymbol{b}_g, \boldsymbol{b}_x \in \mathbb{R}^{d_{out}}$ are trainable weights.

The result shows that higher dimension number have better performance when the dimension number is lower certain threshold, however, when the number of dimensionality is greater than the threshold, larger number of parameter and higher dimensionality doesn't contribute to performance. In the case of SNLI, due to its simplicity in language pattern, 250D would be suffice to obtain a good performance. On the other hand, it requires 350D to achieve a competitive performance on MultiNLI. We fail to reproduce our best performance with the new structure on MultiNLI. It shows that the additional layer on highway network doesn't helps convergence.

| Dimension | Param Count | SNLI | Dev Accuracy Matched | Mismatched |
|---|---|---|---|---|
| 1. DIIN(448) | 4.36 M | 88.4 | 79.2 | 79.1 |
| 2. 10 | 708 K | 81.6 | 71.7 | 71.9 |
| 3. 30 | 765 K | 85.2 | 75.0 | 74.9 |
| 4. 50 | 832 K | 86.0 | 76.1 | 76.7 |
| 5. 100 | 1.05 M | 86.9 | 76.9 | 77.1 |
| 6. 150 | 1.34 M | 87.6 | 77.6 | 77.4 |
| 7. 250 | 2.14 M | 88.1 | 78.1 | 77.7 |
| 8. 350 | 3.23 M | 88.0 | 78.7 | 78.2 |
| 9. 447 | 4.55 M | 88.1 | 78.4 | 78.0 |
| 10. 540 | 6.08 M | 88.1 | 78.8 | 78.7 |
| 11. 600 | 7.20 M | 88.4 | 78.7 | 78.2 |

Table 7: Dimensionality and parameter number study result.

