# OpenReview forum: "Natural Language Inference over Interaction Space"
_ICLR.cc/2018/Conference — Accept (Poster)_

### Official Review · AnonReviewer1 · 2017-11-26
**Does interaction tensor contains the required information?**

**Rating:** 6
**Confidence:** 4

**Review:**

Thank you for this paper! It is very nice piece of work and the problem of coding the "necessary semantic information required for understanding the text" is really a very important one.

Yet, as many papers, it fails to be clear in describing what is its real novelty and the introduction does not help in focussing what is this innovation.

The key point of the paper seems to demonstrate that the "interaction tensor contains the necessary semantic information required for understanding the text". This is a clear issue as this demostration is given only using 1) ablation studies removing gates and non capabilities; 2) analyzing the behavior of the model in the annotated subpart of the MultiNLI corpus; 3) a visual representation of the alignment produced by the model. Hence, there is not a direct analysis of what's inside the interaction tensors. This is the major limitation of the study. According to this analysis, DIIN seems to be a very good paraphrase detector and word aligner. In fact, Table 6 reports the astonishing 100% in paraphrase detection for the Mismatch examples. It seems also that examples where rules are necessary are not correctly modeled by DIIN: this is shown by the poor result on Conditional and Active Passive. Hence, DIIN seems not to be able to capture rules.

For a better demostration, there should be a clearer analysis of these "interaction tensors". The issue of the interpretability of what is in these tensors is gaining attention and should be taken into consideration if the main claim of the paper is that: "interaction tensor contains the necessary semantic information required for understanding the text". Some interesting attempts have been made in "Harnessing Deep Neural Networks with Logic Rules", ACL 2016 and in "Can we explain natural language inference decisions taken with neural networks? Inference rules in distributed representations", IJCNN 2017.


Minor issues
======
Capital letters are used in the middle of some sentences, e.g. "On the other hand, A mul",  "powerful capability, We hypothesize"

---

> ### Author Response · Authors · 2018-01-05
> **Thanks for your insightful review.**
>
> We thank reviewer R1 for insightful comments and feedback. We have updated our paper to clarify the noted defects and we list our responses as follows:
>
> Novelty of our approaches?
> The novelty of our approaches stands on two sides: performance side and method side.
> On performance side, our approach pushed the new state-of-the-art performance on various dataset which is a strong indicator of the novelty.
> On method side, our approach learned deep semantic feature from interaction space. To the best of our knowledge, it is the first attempt to solve NLI tasks with such approach.
>
>
> Main claim?
> We have updated the main claim of the paper to “the attention weight can assist with machine to understanding the text”. As noted in the review, our system is a very good paraphrase detector and word aligner. It is because our system is an exploration of attention mechanism and the feature extractor can well extract the alignment feature from attention weight (the interaction tensor). Given the fact that attention is not designed for capturing rules, our system is not strong on the samples with rules involved. How to solve the ruled based samples will be the focus of our future work.
>
>
> Model interpretability?
> The interaction tensor is essentially a kind of high-order attention weight to model the alignment of two sentences in a more flexible way. As compared between experiment 1 and 8 in ablation study, a high-order attention weight outperform the regular attention weight with a large margin.
> The deep learning research is known to be challenging in interpreting the model. In NLP deep learning research, we often empirically evaluate the model capacity with their performance on challenging task.
> Even though it is challenging to interpret our models directly, we tried to implement ablation studies, error analysis and visual representation to provide an intuition for our approach.
> We thank R1 for providing very interesting paper.
>
>
> Grammer issues?
> Thanks for pointing it out. We have updated the paper to remove the noted defects.

---

### Official Review · AnonReviewer2 · 2017-11-26
**The result of this paper for recognizing textual entailment is very interesting; however, the paper is not well-written and it doesn't provide motivation and intuition for each component of their model and it needs major revision.**

**Rating:** 5
**Confidence:** 5

**Review:**

This paper proposes Densely Interactive Inference Network to solve recognizing textual entailment via extracting a semantic feature from interaction tensor end-to-end. Their results show that this model has better performance than others.

Even though the results of this paper is interesting, I have the problem with paper writing and motivation for their architecture:

- Paper pages are well beyond 8-page limits for ICLR. The paper should be 8-pages + References. This paper has 11 pages excluding the references.
-  The introduction text in the 2nd page doesn't have smooth flow and sometimes hard to follow.
-  In my view section, 3.1 is redundant and text in section 3.2 can be improved
-  Encoding layer in section 3.2 is really hard to follow in regards to equations and naming e.g p_{itr att} and why choose \alpha(a,b,w)?
-  Encoding layer in section 3.2, there is no motivation why it needs to use fuse gate.
-  Feature Extraction Layer is very confusing again. What is FSDR or TSDR?
-  Why the paper uses Eq. 8? the intuition behind it?
-  One important thing which is missing in this paper, I didn't understand what is the motivation behind using each of these components? and how each of these components is selected?
- How long does it take to train this network? Since it needs to works with other models (GLOV+ char features + POS tagging,..), it requires lots of effort to set up this network.

Even though the paper outperforms others, it would be useful to the community by providing the motivation and intuition why each of these components was chosen. This is important especially for this paper because each layer of their architecture uses multiple components, i.e. embedding layer [Glov+ Character Features + Syntactical features].  In my view, having just good results are not enough and will not guarantee a publication in ICLR, the paper should be well-written and well-motivated in order to be useful for the future research and the other researchers.
In summary, I don't think the paper is ready yet and it needs significant revision.



---------------------------------------------------------------------------------------------------------------------------------------------------------------
---------------------------------------------------------------------------------------------------------------------------------------------------------------
Comments after the rebuttal and revision :
I'd like thanks the authors for the revision and their answers.
Here are my comments after reading the revised version and considering the rebuttal:
- It is fair to say that the paper presentation is much better now. That said I am still having issues with 11 pages.
- The authors imply on page 2, end of paragraph 5,  that this is the first work that shows attention weight contains rich semantic and previous works are used attention merely as a medium for alignment.  Referring to the some of the related works (cited in this paper), I am not sure this is a correct statement.
- The authors claim to introduce a new class of architectures for NLI and generability of for this problem. In my view, this is a very strong statement and unsupported in the paper, especially considering ablation studies (table 5). In order for the model to show the best performance, all these components should come together. I am not sure why this method can be considered a class of architecture and why not just a new model?

some other comments:
- In page 4, the citation is missing for highway networks
- Page 5, equation 1, the parenthesis should close after \hat{P}_j.

Since the new version has been improved, I have increased my review score.  However, I'm still not convinced that this paper would be a good fit at ICLR given novelty and contribution.

---

> ### Author Response · Authors · 2018-01-05
> **Thanks for your insightful review.**
>
> We thank reviewer R2 for insightful comments and feedback. We have updated our paper to clarify the noted defects and we list our responses as follows:
>
> Beyond 8-page limit?
> 8-page length is a recommendation rather than a restriction in ICLR 2018. We have done extensive experiments to show the strength of our model and wish to be as comprehensive as possible. In order to keep the paper length short, we’ll move part of content to supplementary materials.
>
> Introduction in the 2nd page hard to follow?
> Thanks for pointing it out. We have reworded the introduction in second page to make it clear.
>
> Is section 3.1 redundant?
> The section 3.1 serves the purpose of introducing a general framework on the NLI task. As indicated by paper “Supervised Learning of Universal Sentence Representations from Natural Language Inference Data”, there is a generic NLI training scheme. We intend to propose a new training scheme. The experiment 3 in the ablation study intends to compare between these two approaches.
>
> Bad equation notations?
> Thanks for pointing it out. We have updated the paper to clarify the equations.
>
> Why choosing \alpha(a,b) and \beta(a,b)?
> We choose this notation on purpose because both of them are the function of attention. Attention has various kinds of implementation and we want to keep the option open. We have attempted multiple version of attention and they perform similarly. Therefore, we only choose one version to be presented in this paper. Other implementations of attention mechanism are beyond the discuss of this paper.
>
> The motivation of fuse gate?
> As indicated in encoding layer description as well as in ablation study experiment 6&7, we are motivated to use fuse gate as a skip connection. Even though using addition as skip-connection is useful to combine the output of self-attention and highway network, we empirically demonstrate that using fuse gate, which weights the new information and old one to carefully fuse them, is better than using addition. Similar observation can be found in paper “Ruminating Reader: Reasoning with Gated Multi-Hop Attention”.
>
> What is FSDR or TSDR?
> Thanks for pointing it out. FSDR stands for “first scale down ratio” and TSDR stands for “transition scale down ratio”. They are two scale down ratio hyperparameters in feature extraction layer. In the updated version of paper, we have renamed these two hyperparameters with greek characters \eta and \theta.
>
>
> Why the paper uses Eq. 8? The intuition?
> Equation is an auxiliary function that helps L2 regularization. As the training goes by, the model starts to overfit the training set. Therefore, we wish to increase the L2 regularization strength along with the training process.
>
> How long does it take to train this network?
> The training time depends on the task. With a single TITAN X GPU, it takes near 48 hours to converge on MultiNLI, 24 hours on SNLI and Quora Question Pair corpora. Though our model has many components as input, they only need to be processed once. The setup time is ignorable after the dataset has been processed. We have open sourced our code and processed dataset. However, due to the anonymous policy, we intend to post the link here after the paper is accepted.
>
> The motivation behind complex component e.g. embedding layer?
> Thanks for pointing it out. We have updated the paper to make sure it is as comprehensive as possible. If a component is one of the common practices, then we have cited the related references. If a component is innovated, then we have explained the motivation in text and empirically evaluate it with ablation studies.

---

> > ### Author Response · Authors · 2018-01-20
> > **Response for "post-rebuttal review"**
> >
> > Over claim about novelty?
> > After doing extensive research and consulting various experts, we are 100% certain that no previous work have taken similar approach on recognising textual entailment task before. The related work section only introduces the relevant literatures that adopts attention or CNN modules. If there are works which have shown the point we are arguing, we eager to see it.
> >
> > Necessity of formulate the model into a new framework?
> > The section 3.1 provides intuition and motivation how and why each module works together. By having the ablation study, we empirically assess the necessity of each module. By reducing 20% of the previous state-of-the-art performance on MultiNLI corpus, we show the power of this kind of new architecture. To encourage researchers to further working on this direction, we feel it is necessary to formulate it into a general framework.

---

### Official Review · AnonReviewer3 · 2017-11-27
**The paper explores interaction tensors (or attention weights) to capture semantic information to help solve NLI or NLI alike tasks, but contributions are not clear as the claims are not convincingly supported by experiments.**

**Rating:** 6
**Confidence:** 4

**Review:**

Pros:
The paper proposes a “Densely Interactive Inference Network (DIIN)” for NLI or NLI alike tasks. Although using tensors to capture high-order interaction and performing dimension reduction over that are both not novel, the paper explores them for NLI. The paper is written clearly and is very easy to follow. The ablation experiments in Table 5 give a good level of details to help observe different components' effectiveness.
Cons:
1) The differences of performances between the proposed model and the previous models are not very clear. With regard to MultiNLI, since the previous results (e.g., those in Table 2) did not use cross-sentence attention and had to represent a premise or a hypothesis as a *fixed-length* vector, is it fair to compare DIIN with them? Note that the proposed DIIN model does represent a premise or a hypothesis by variable lengths (see interaction layer in Figure 1), and tensors provide some sorts of attention between them. Can this (Table 2) really shows the advantage of the proposed models? However, when a variable-length representation is allowed (see Table 3 on SNLI), the advantage of the model is also not observed, with no improvement as a single model (compared with ESIM) and being almost same as previous models (e.g., model 18 in Table 3) in ensembling.
2) Method-wise, as discussed above, using tensors to capture high-order interaction and performing dimension reduction over that are both not novel.
3) The paper mentions the use of untied parameters for premise and hypothesis, but it doesn’t compare it with tied version in the experiment section.
4) In Table 6, for CONDITIONAL tag, why the baseline models (lower total accuracies) have a 100% accuracy, but DIIN only has about a 60% accuracy?

---

> ### Author Response · Authors · 2018-01-05
> **Thanks for your insightful review**
>
> We thank reviewer R3 for insightful comments and feedback. We have updated our paper to clarify the noted issues and we list our responses as follows:
>
> Novelty of our approaches?
> The novelty of our approaches stands on two sides: the performance side and the method side.
> On performance side, our approach pushed the new state-of-the-art performance on various dataset which is a strong indicator of the novelty. The model reduces 20% of error rate of previous state-of-the-art performance on MultiNLI corpus.
> On method side, our approach learned deep semantic feature from interaction space. To the best of our knowledge, it is the first attempt to solve NLI tasks with such approach.
>
> Performance difference between proposed model and previous models?
> The intention of this paper is to explore a new approach that extracting deep semantic features from a dense alignment tensor to solve NLI tasks. The experiment results show that our approach achieves and pushes state-of-the-art performance on multiple dataset thus justifying its potential. In error analysis section, we show the strength of our model on samples with paraphrase, antonyms and overlapping words. On the other hand, the model has limitation on samples with rules (CONDITIONAL tags). Therefore, our approaches and other approaches is in complementary with each other.
>
> Why the model has lower accuracy on samples with CONDITIONAL tag (rules)?
> Since our approach is mainly based on attention alignment tensor, our approach has advantage over cases such as paraphrase where attention mechanism excels. How to solve samples with rules will be the focus of our future work.
>
> Unfair comparison with other model in MultiNLI?
> Though other models are subject to the limitation of RepEval 2017 workshop, ESIM baseline provided by Williams et al (2017) are not subject to such restrict. ESIM shows state-of-the-art performance on SNLI corpus. Our model outperforms their model on MultiNLI with near 6.5% of accuracy on single model setting. It is a fair comparison and our model has clear advantage over it.
>
> No improvement over SNLI?
> As stated by Williams et al (2017), SNLI mainly uses caption sentences which is relatively simple in syntax and semantics. However, the MultiNLI corpus, a follow-up corpus on SNLI, collects sentences from various kinds of literature and is far more challenging. Even though our model has similar performance with ESIM on SNLI, it outperforms ESIM with a large margin on MultiNLI corpus.
>
> Missing of untied parameter ablation study?
> Thanks for pointing it out. We have included the study and result into the updated version of paper. As a reference, the tied parameter version of DIIN achieves 78.5 on matched-dev set and 78.3 on mismatched-dev set.
>
> Reference:
> Williams, A., Nangia, N., & Bowman, S. R. (2017). A Broad-Coverage Challenge Corpus for Sentence Understanding through Inference. arXiv.org.

---

### Public Comment · ~Martin_Mirakyan1 · 2017-12-13
**ICLR Reproducibility Challenge - Feedback and several Questions**

Thank you for the detailed description of the architecture. I choose this paper for ICLR Reproducibility Challenge and I think I could reproduce the main parts of the architecture. Still, there are a few details that I wasn't able to understand from the paper:
1. How are the weights of the network initialized?
2. What is the length of character input? As far as I understood for every sentence in the given dataset we take a hard cutoff of words (32 words for SNLI, etc) and for character-embedding input we need to take some number X and pad words shorter than X with 0s or cut longer ones. So how is that number X determined?
3. What is the size of the character embedding and kernel-size of the convolution applied on character embeddings?
4. Have you tried to experiment with embeddings of Syntactical features instead of taking their one-hot representations?

It was easy to follow the paper as it was well organized with clearly stated details and explanations.
Here is the repository where I've tried to reproduce the architecture and results stated in this paper: https://github.com/YerevaNN/DIIN-in-Keras

Here is the link to our reproducibility report: https://arxiv.org/abs/1802.03198

---

> ### Author Response · Authors · 2018-01-05
> **Thanks for your interests. Here is my response.**
>
> Thank you Martin, I do appreciate your interest in our work.
> Though we have released the code, due to the anonymous policy I can't provide the link here. Therefore, I'll provide the link to you after the paper is accepted/rejected.
>
> 1. Except the embedding, all weights are randomly initialised with Tensorflow's default setting.
> 2. The length of character input is 16 for each token. During preprocessing we need to pad or cut to satisfy the constraint. We have updated the paper to make this implementation detail more clear.
> 3. The size of character embedding is 100 and the kernel-size of the 1D convolution on character embedding is 5. We have updated the paper to make this implementation detail more clear.
> 4. It is a good point to replace one-hot representation of syntactical feature with their embedding. However,  we didn't observe any improvement over our baseline empirically. The embedding serves the purpose of dimension reduction. A linear layer over one-hot input will behave the same way as embedding does. Since the dimensionality of syntactical feature is very small (48D), we believe it is not necessary to replace it.
>
> Your code is neat in general. I hope my code will give you additional insight in near future.

---

### Author Response · Authors · 2018-01-05
**Update notes**

Here we list all the revisions since submission:

As of Jan. 5 2018:
1. We reworded the introduction to clarify our novelty and to make the sentences more readable.
2. We updated section 3.2 with clearer motivation of each component.
3. We cleaned the math notation in section 3.2.
4. We included an extra ablation experiment to indicate how tied encoding weight helps the performance.
5. We added the character feature implementation detail.
6. We reserved a placeholder for code link in section 3.2

---

### Public Comment · ~Martin_Mirakyan1 · 2018-02-12
**Reproducibility Report**

Thanks a lot for the detailed paper and insightful comments. We continued our experiments after the authors answered our questions and got better results after tuning the hyperparameters and network architecture.

More details about our experiments can be found in the report: https://arxiv.org/abs/1802.03198

---

> ### Author Response · Authors · 2018-02-12
> **Thank you**
>
> Thank you Martin. It is a very comprehensive report.

---

### Decision · Program_Chairs · 2018-01-29
**ICLR 2018 Conference Acceptance Decision**

**Decision:**

Accept (Poster)

**Comment:**

This paper presents a marginally interesting idea -- that of an interaction tensor that compares two sentence representations word by word, and feeds the interaction tensor into a higher level feature extraction mechanism.  It produces good results on multi-NLI and SNLI datasets.  There is some criticism about comparing with several baselines for multi-NLI where there was a restriction of not using inter-sentence comparison networks, but the authors do compare with a similar approach without that restriction and shows improvements.   However, there is no solid error analysis that shows what type of examples this interaction tensor idea captures better than other strong baselines such as ESIM. Overall, the committee feels this paper will add value to the conference.